# The Expression and Function of Metastases Associated Lung Adenocarcinoma Transcript-1 Long Non-Coding RNA in Subchondral Bone and Osteoblasts from Patients with Osteoarthritis

**DOI:** 10.3390/cells10040786

**Published:** 2021-04-01

**Authors:** Fawzeyah A. Alnajjar, Archana Sharma-Oates, Susanne N. Wijesinghe, Hussein Farah, Dominika E. Nanus, Tom Nicholson, Edward T. Davis, Simon W. Jones

**Affiliations:** 1Institute of Inflammation and Ageing, MRC-ARUK Centre for Musculoskeletal Ageing Research, University of Birmingham, Birmingham B15 2TT, UK; fxa790@student.bham.ac.uk (F.A.A.); a.sharma-oates@bham.ac.uk (A.S.-O.); s.n.wijesinghe@bham.ac.uk (S.N.W.); hxf820@student.bham.ac.uk (H.F.); d.nanus@bham.ac.uk (D.E.N.); t.a.nicholson@bham.ac.uk (T.N.); 2The Royal Orthopaedic Hospital, Birmingham B31 2AP, UK; edward.davis@nhs.net

**Keywords:** osteoarthritis, osteoblasts, MALAT1, long non-coding RNA

## Abstract

Metastasis Associated Lung Adenocarcinoma Transcript-1 (MALAT1) is implicated in regulating the inflammatory response and in the pathology of several chronic inflammatory diseases, including osteoarthritis (OA). The purpose of this study was to examine the relationship between OA subchondral bone expression of MALAT1 with parameters of joint health and biomarkers of joint inflammation, and to determine its functional role in human OA osteoblasts. Subchondral bone and blood were collected from hip and knee OA patients (*n* = 17) and bone only from neck of femur fracture patients (*n* = 6) undergoing joint replacement surgery. Cytokines were determined by multiplex assays and ELISA, and gene expression by qPCR. MALAT1 loss of function was performed in OA patient osteoblasts using locked nucleic acids. The osteoblast transcriptome was analysed by RNASeq and pathway analysis. Bone expression of MALAT1 positively correlated to serum DKK1 and galectin-1 concentrations, and in OA patient osteoblasts was induced in response to IL-1β stimulation. Osteoblasts depleted of MALAT1 exhibited differential expression (>1.5 fold change) of 155 genes, including PTGS2. Both basal and IL-1β-mediated PGE2 secretion was greater in MALAT1 depleted osteoblasts. The induction of MALAT1 in human OA osteoblasts upon inflammatory challenge and its modulation of PGE2 production suggests that MALAT1 may play a role in regulating inflammation in OA subchondral bone.

## 1. Introduction

Osteoarthritis (OA) is a degenerative joint disease and a leading cause of pain and disability for which there are currently no approved pharmacological disease modifying therapeutics [1]. Current OA treatment is limited to generic analgesics that have limited efficacy, physical therapy and ultimately joint replacement surgery [2,3]. Non-pharmacological therapeutic strategies such as platelet rich plasma and mesenchymal stem cells, aimed at promoting cartilage repair, hold promise but require further clinical studies to confirm efficacy and safety [4]. There is therefore a high unmet healthcare need to better understand the pathophysiology of OA and to develop an effective disease modifying therapeutic [5]. Importantly, despite initially considered to be a disease solely of the articular cartilage, it is now widely accepted that OA is a disease that encompasses the whole joint, including underlying subchondral bone tissue [2]. Pathological changes to OA subchondral bone tissue, characterised by trabecular thickening [6], abnormal type I Collagen production [7,8] and the formation of osteophytes, occurs early in the disease course of OA [9] and involves changes to the osteoblast phenotype [10]. Furthermore, in animals that are prone to developing OA trabecular thickening occurs prior to cartilage degeneration [11] and thus it has been suggested that this alteration to the bone architecture may pathologically alter the loading biomechanics, thus promoting cartilage degeneration [12]. Therefore, identifying novel regulators that mediate the functional phenotype of osteoblasts, the cells which regulate bone remodelling, may identify new opportunities for the development of new OA therapeutics that target the bone.

Recently, long non-coding RNAs (lncRNAs) have emerged as novel epigenetic regulators of gene transcription [13,14] and of mediating several cellular processes including cellular proliferation, cell cycle control, apoptosis and the innate inflammatory response [15,16,17,18]. As such, lncRNAs have now been implicated in the pathology of several chronic inflammatory diseases [19,20], including OA [21]. Indeed, lncRNAs have been identified that are differentially expressed in OA diseased cartilage and which mediate the IL-1β inflammatory response in OA chondrocytes [21,22]. Furthermore, we recently reported that Metastasis Associated Lung Adenocarcinoma Transcript-1 (*MALAT1*) lncRNA modulates the inflammatory phenotype of synovial fibroblasts in the OA synovial joint lining by mediating the production of CXCL8 [23]. However, importantly, *MALAT1* lncRNA has now emerged as a central mediator of osteoblast function and bone homeostasis. Expression of *MALAT1* is reported to be greater in the bone tissue of patients who exhibit aseptic loosening following a hip replacement [24] and its upregulation has been implicated in lumber intervertebral disc degeneration [25]. In vitro, knockdown of *MALAT1* has been demonstrated to inhibit the proliferation of the human osteoblast cell line hFOB 1.19 [24], and *MALAT1* sponging of the microRNA miR-30 has been shown to promote the osteoblast differentiation of mesenchymal stem cells by inducing *RUNX2* expression [26]. Furthermore, *MALAT1* is associated with abnormal osteogenic and adipogenic differentiation of BMSCs in the patients with osteonecrosis of the femoral head [27].

These data suggest that *MALAT1* could play a key role in the pathological changes that occur in OA diseased bone. However, currently the expression and functional role of *MALAT1* in OA subchondral bone has not been reported. Therefore, the aim of this study was to profile the expression of *MALAT1* in the subchondral bone tissue of patients with either knee or hip OA and its relationship to parameters of joint damage and to examine the functional role of *MALAT1* in OA patient primary osteoblasts.

## 2. Materials and Methods

### 2.1. Patient Recruitment

Following ethical approval (UK NRES 16/SS/0172 & NRES 14/ES/1044 approved on 31 October 2016 and 12 August 2014, respectively), subchondral bone tissue and blood was collected from a total of 17 patients (9 males, 8 females) with end-stage OA comprising of 9 patients with hip OA and 8 patients with knee OA and from neck of femur fracture (NOF#) patients without OA (*n* = 6), who were undergoing surgery at Russell’s Hall Hospital (Dudley, UK) or the Royal Orthopaedic Hospital (Birmingham UK). OA patient characteristics are detailed in Table 1. All participants demonstrating secondary causes of OA, such as avascular necrosis, Perthes disease, developmental dysplasia, previous acetabular or femoral neck fractures and slipped upper femoral epiphysis, were excluded from the study. Informed consent was obtained from all subjects involved in the study.

### 2.2. Isolation and Culture of Primary Osteoblasts

Primary OA osteoblasts were cultured from OA subchondral bone tissue. The bone was cut into small pieces approximately 2 mm^3^ and washed 3 times in 30 mL of complete osteoblast culture media (DMEM, 10% FBS, 100 Units/mL Penicillin, 100 ug/mL Streptomycin, 1% Non-essential amino acids, L-glutamine 2 mM, ascorbic acid 50 ug/mL, Dexamethasone 10 nM, and β-glycerophosphate 2 mM) to remove excess fat. The bone chips were then placed in a T75 flasks with fresh culture media and incubated in at 37 °C in a humidified atmosphere containing 5% CO_2_. The media was changed every 3–4 days, and the chips were removed after 7–14 days when the osteoblasts outgrowth occurred.

### 2.3. Analysis of Serum Cytokines

The serum concentration of 24 cytokines were determined using Luminex multiplex platform (Luminex R&D systems) according to the manufacturer’s instructions, having been diluted to 1:2 in assay buffer.

### 2.4. MALAT1 Functional Studies in Human OA Osteoblasts

In vitro loss of function studies were performed using lipofectamine 3000 (Qiagen, Manchester, UK) to transfect primary osteoblasts with two different Locked nucleic acids (LNAs) targeting *MALAT1* or with a control (NC) LNA. To determine the effect of *MALAT1* knockdown on the OA osteoblast transcriptome, total RNA was extracted (RNeasy columns, Qiagen Manchester UK) 24 h following transfection with LNAs. RNA integrity (RIN) was evaluated (Agilent Bioanalyser, Cheshire UK) with a RIN of >7 deemed of sufficient quality for RNA sequencing analysis using the QuantSeq 3′ kit (Lexogen, Vienna, Austria). The sequenced reads were mapped to the hg38 reference human genome using Star Aligner version 2.5.2b [28]. Differential gene expression analysis and log2 fold changes were computed using DESeq2 version 1.26.0 [29].

The effect of *MALAT1* knockdown on the production of PGE2 and osteoprotegrin (OPG) was determined by ELISA according to the manufacturer’s instructions (RnD systems, Oxford, UK). Alkaline phosphatase (ALP) activity was quantified in osteoblast lysates prepared in RIPA buffer diluted 1:5 with 1 mM MgCl2. In brief, diluted osteoblast lysates were combined with ALP substrate (Sigma, Gillingham, UK) and incubated at 37 °C for 15 min. The reaction was stopped with the addition of 0.1 N NaOH and absorbance read at 405 nm on a microplate reader. The degree of osteoblast mineralisation was quantified by staining of mineralised nodules using an alizarin red solution (0.5% Alizarin Red in 1% ammonia hydroxide at pH 4.5; Sigma, Gillingham, UK). Following 10 min incubation at room temperature, cells were washed with PBS to remove excess stain, and then incubated in 10% cetyl pyridinium chloride (Sigma, Gillingham, UK) for 10 min. The supernatant was then collected, diluted 1:10 with the 10% cetyl pyridinium chloride and absorbance read at OD550 nm on a microplate reader.

### 2.5. Pathway Analysis

Pathway analysis was performed using the software Ingenuity Pathway Analysis (IPA; www.ingenuity.com, accessed on 31 January 2020). Differentially expressed genes (fold change of ± >1.5, *p* < 0.05) were analysed using a core functional analysis to identify significant canonical pathways and cellular processes. 

### 2.6. Statistical Analysis

All statistics were performed using Graph Pad Prism 9.0. ANOVA followed by post hoc analysis of Bonferroni was used to test statistical significance where appropriate. Pearson correlations were performed to test the strength of association between two variables. *p*-value < 0.05 was considered as significant.

## 3. Results

### 3.1. Expression of MALAT1 in OA Subchondral Bone and Relationship to Patient Characteristics and OA Disease Severity

*MALAT1* was highly expressed in OA subchondral bone tissue, with on average greater expression in hip OA compared to non-OA hip (NOF#). However, this did not reach significance and there was no significant difference in expression between patients with knee OA and those with hip OA (Figure 1A). Furthermore, although on average there was greater *MALAT1* expression in OA patients with greater disease severity (KL4 and joint space < 1 mm), compared to those patients with KL grade 3 and joint space > 1 mm, there was no significant relationship between *MALAT1* expression and disease severity (Figure 1B; Appendix A).

Next, we examined whether the subchondral bone expression of *MALAT1* was related to biomarkers of inflammation by measuring the concentration of a panel of 22 cytokines in OA patient serum by Luminex. There were significant positive correlations between *MALAT1* expression and the serum concentration of both the Wnt pathway inhibitor DKK1 (r^2^ = 0.3, *p* = 0.04) and galectin-1 (r^2^ = 0.44, *p* = 0.009) (Figure 1C; Table 2). However, it should be noted that these findings were not significant upon correcting for multiple comparisons (FDR), and therefore a type I error cannot be ruled out. 

### 3.2. MALAT Expression in OA Osteoblasts Is Induced by Inflammatory Challenge

Next, using an in vitro model of primary OA patient osteoblasts, we examined whether the expression of *MALAT1* was induced upon an inflammatory challenge. To this end, primary osteoblasts from *n* = 3 OA patients were stimulated for either 6 h or 24 h with IL-1β (1 ng/mL), and the expression of *IL-6* and *MALAT1* determined by qPCR. As expected, stimulation of osteoblasts with IL-1β induced a significant increase in the expression of the pro-inflammatory cytokine *IL-6* within 24 h (23-fold, *p* < 0.05), compared to non-stimulated control. Similarly, osteoblasts stimulated with IL-1β for 24 h exhibited significant increase in the expression of *MALAT1* (4-fold, *p* < 0.05), compared to non-stimulated control (Figure 1D).

### 3.3. The Effect of MALAT1 Knockdown on the OA Osteoblast Transcriptome

To examine the functional role of *MALAT1* in OA subchondral osteoblasts we next conducted loss of function studies by performing targeted knockdown of *MALAT1* expression using locked nucleic acids (LNAs) and examined the resulting osteoblast transcriptome by RNA sequencing. Primary OA osteoblasts (*n* = 3 OA patients) were transfected with either a non-targeting control LNA or one of two LNA duplexes targeting *MALAT1*. Following 24 h transfection, LNAs targeting *MALAT1* induced knockdown of between 60–90% in the expression of *MALAT1*, compared to the control LNA (Figure 2A). RNAseq identified 155 transcripts (82 upregulated and 73 downregulated), which were differentially expressed (>1.5-fold change, *p*-value ≤ 0.05) in osteoblasts transfected with the *MALAT1* LNAs, compared to LNA control transfected cells. Of the upregulated transcripts, 80 were protein coding genes (including *PTGS2*) and 2 were antisense lncRNAs. Of the down-regulated transcripts, 66 were protein-coding genes and 7 were lncRNAs, which comprised of 2 lincRNAs, 3 antisense, 1 sense intronic lncRNA and 1 pseudogene (Figure 2B,C, Appendix A).

To understand the functional effect of this *MALAT1* mediated change in osteoblast transcriptome we next analysed the 155 differentially expressed genes by bioinformatic pathway analysis (Ingenuity Pathway Analysis) to identify significantly affected canonical pathways and cellular processes. The most significant canonical pathways affected included phosphatidylcholine biosynthesis, fMLP signalling in neutrophils, NAD biosynthesis, eicosanoid biosynthesis and prostanoid biosynthesis. The significantly affected cellular processes included cell–cell signalling, DNA replication, cellular growth and proliferation and cellular development (Figure 2D).

### 3.4. Modulation of MALAT1 Expression in Osteoblasts Induced PGE2 Secretion but Did Not Impair Osteoblast OPG Secretion or Their Innate Ability to Mineralise

Given the finding that knockdown of *MALAT1* induced the expression of *PTGS2*, we next examined if this was reflected in a greater production of the prostaglandin PGE2, a purported mediator of pain and inflammation in OA. To this end, OA osteoblasts were transfected with either a control LNA or one of two *MALAT1* LNAs for 24 h. Cells were then either stimulated for 24 h with IL-1β (1 ng/mL) or left unstimulated and secretion of PGE2 quantified by ELISA. *MALAT1* knockdown was confirmed by qPCR (Figure 3A). Compared to LNA control, cells depleted of *MALAT1* exhibited greater basal and IL-1β induced PGE2 production (2-fold, *p* < 0.05) (Figure 3B).

Next, we investigated whether chronic loss of function of *MALAT1* would affect the innate function of osteoblasts with regard to bone remodelling by determining OPG production, alkaline phosphatase activity and osteoblast mineralisation. To this end, OA osteoblasts were transfected with either a control LNA or a *MALAT1* LNA twice per week for a period of 3 weeks. During the 3-week time course, supernatants were collected to analyse OPG production by ELISA and cells lysed to measure alkaline phosphatase activity. At the end of the 3 weeks, cells were stained with alizarin red to quantify mineralisation. Twice weekly transfection with LNA maintained the significant knockdown in *MALAT1* expression of between 65–85% (Figure 3C). Production of OPG from both control and *MALAT1* LNA transfected cells increased throughout the 3-week time course. However, there was no significant difference in either the rate of OPG production or in the total amount secreted (Figure 3D). Furthermore, there was no significant difference in either alkaline phosphatase activity (Figure 3E) or the degree of osteoblast mineralisation (Figure 3F) between *MALAT1* depleted osteoblasts and control cells.

## 4. Discussion

This paper reports the expression and functional role of the long non-coding RNA *MALAT1* in OA patient subchondral bone tissue and primary OA osteoblasts. *MALAT1* was found to be highly expressed (compared to non-OA hip) in the subchondral bone tissue of both knee and hip OA patients irrespective of disease severity. On average *MALAT1* expression was greater in the bone tissue from patients with greater OA severity, as measured by either KL grade or joint space narrowing. However, it should be noted that all the OA patients in this study had advanced end-stage disease, with radiographic severity determined to be either KL3 or KL4, and most patients having <1 mm joint space. Therefore, we cannot comment on whether expression of *MALAT1* would differ in the subchondral bone of patients with early OA. 

Similar to previous findings on the role of *MALAT1* in OA synovial fibroblasts [23], *MALAT1* expression in OA osteoblasts was induced during the IL-1β inflammatory response. This suggests *MALAT1* may play an important regulatory role in bone homeostasis under the inflammatory conditions exhibited in OA patients. Of note, although the subchondral bone expression of *MALAT1* was not related to the serum concentration of IL-1β we did observe positive correlations between *MALAT1* expression and the serum concentration of both DKK1 and galectin 1. DKK1 is an endogenous inhibitor of the Wnt/beta-catenin signalling pathway and is implicated in bone development and in the pathological remodelling of bone in both OA and osteoporosis and mediating inflammation-induced bone loss by inhibiting osteoblast differentiation [30,31]. In osteoporosis patients, serum levels of DKK1 are negatively associated with bone mineral density in the femoral head and lumbar spine [32]. In contrast, galectin-1 has been identified as secreted protein which promotes osteoblast differentiation [33], and in mediating osteoclast activity in osteolytic bone disease [34]. 

LNA-mediated knockdown of *MALAT1* in primary human OA osteoblasts profoundly affected the transcriptomic phenotype with pathway analysis revealing significant activation of pathways that promote the production of inflammatory prostacyclins and eicosanoids. Amongst the most differentially expressed upregulated genes were *TNFSF12* (>11-fold upregulated), which encodes TWEAK (TNF-related weak inducer of apoptosis). TWEAK is a known mediator of inflammatory bone remodelling [35], and targeted inhibition of the TWEAK/fibroblast growth factor inducible 14 (FnF14) signalling pathway has been proposed as a therapeutic strategy to reduce bone resorption in rheumatoid arthritis patients [36]. In addition, osteoblasts depleted of *MALAT1* exhibited a significant >10-fold upregulation in the expression of *PTGS2*, the gene which encodes for the enzyme COX2 that mediates the production of inflammatory prostaglandins including the putative OA pain mediator PGE2 [37,38]. Indeed, we found that both basal and IL-1β induced PGE2 secretion was significantly greater in osteoblasts depleted of *MALAT1* compared to control cells. PGE2 acting through the E prostanoid receptors EP2 and EP4 sensitizes nociceptors, possibly by acting synergistically with IL1β to induce *IL-6* and *iNOS* expression [39] and selective COX2 inhibitors such as celecoxib are known to reduce pain and inflammation in OA patients [40]. The subchondral bone is considered to be an important site of OA pain, being innervated with sensory neurones [41] and bone marrow lesions (identified as hyper-intense regions on T1-weighted MRI scans) and bone shape in OA patients being associated with bone pain [42]. Therefore, our finding here that *MALAT1* expression in OA osteoblasts regulates *PTGS2* expression as well as both basal and IL-1β induced PGE2 production suggests that *MALAT1* may play an important role in regulating inflammatory pain in the bone.

Despite the effect of *MALAT1* knockdown on the osteoblast transcriptome and the acute effect on PGE2 production we did not observe any chronic effect of *MALAT1* knockdown on several key osteoblast functions. During a time-period of 3 weeks, we induced sustained *MALAT1* knockdown but did not observe any significant difference in the secretion of OPG or in the activity of ALP and ability of the osteoblast to form mineralised bone nodules. This contrasts with previous publications which have implicated *MALAT1* in mediating both OPG production in osteoblasts [43] as well as ALP activity and mineralisation during osteoblast differentiation [44].

The underlining molecular mechanisms by which lncRNAs mediate their function is complex and for the majority of lncRNAs remains to be determined. Several lncRNAs, including *MALAT1* have been reported to exert a pro-osteogenic function by acting as miRNA sponges [45]. Indeed, the reported role for *MALAT1* in mediating osteoblast differentiation has been linked to the sponging of several miRNAs including miR-204 [45], miR-30 [26] and miR-43 [46]. In our analysis of the OA osteoblast transcriptome analysis following *MALAT1* depletion, we did not identify any miRNAs which were altered upon *MALAT1* KD. However, the isolation of total RNA using columns would likely have excluded many miRNAs from our sequencing analysis.

## 5. Conclusions

In conclusion, the expression of *MALTA1* in OA subchondral bone, its induction in osteoblasts upon inflammatory challenge and its functional role in modulating the production of the prostaglandin PGE2 suggest that *MALAT1* may play an important role in the development of OA bone pain and inflammation.

## Figures and Tables

**Figure 1 cells-10-00786-f001:**
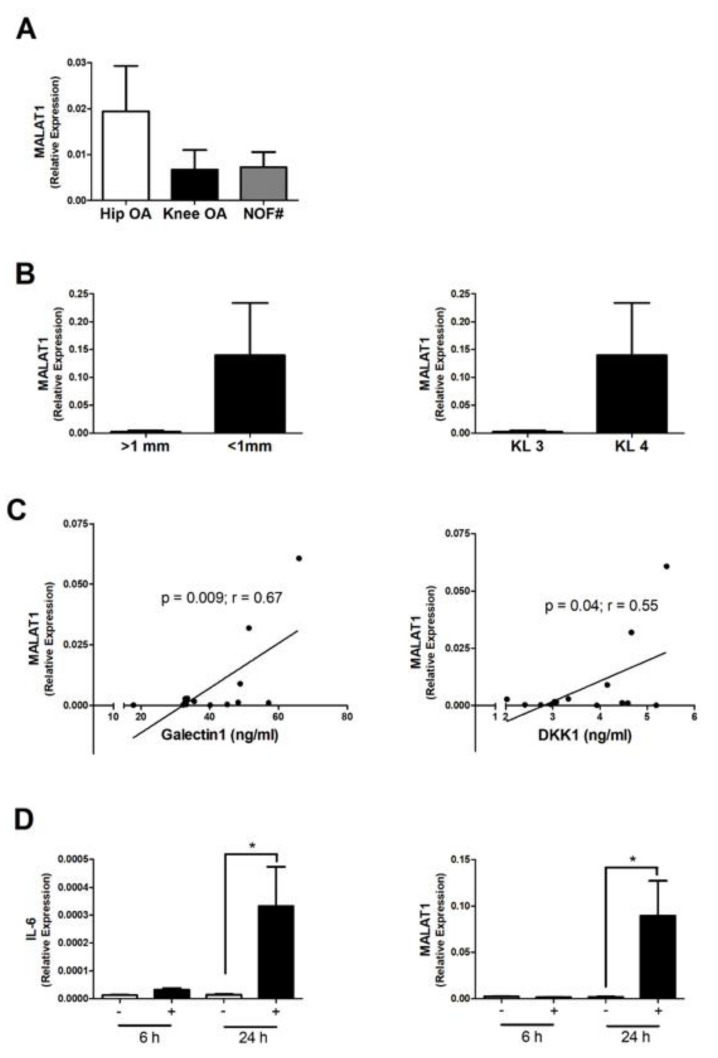
Expression of *MALAT1* in subchondral bone and OA osteoblasts. (**A**) Relative expression of *MALAT1* in the subchondral bone tissue of end-stage hip OA patients (*n* = 9) compared to end-stage knee OA patients (*n* = 8) and non-OA neck of femur fracture (NOF#) patients (*n* = 6). MALAT1 expression was determined by qPCR and normalised to 18S. Bars represent mean expression ± SEM. (**B**) Relative expression of *MALAT1* in the subchondral bone tissue between OA patients with more severe radiographic signs of OA (*n* = 13) with joint space < 1 mm and KL grade 4, compared to those with less severe joint damage (*n* = 4) with joint space > 1 mm and KL grade 3. (**C**) Correlation between subchondral bone expression of *MALAT1* and the serum concentration of Galectin 1 (ng/mL) and DKK1 (ng/mL) in OA patients (*n* = 17). (**D**) Effect of IL-1β stimulation (1 ng/mL) upon the expression of *IL-6* and *MALAT1* in primary OA osteoblasts at 6 h and 24 h. * = *p* < 0.05, compared to non-stimulated control.

**Figure 2 cells-10-00786-f002:**
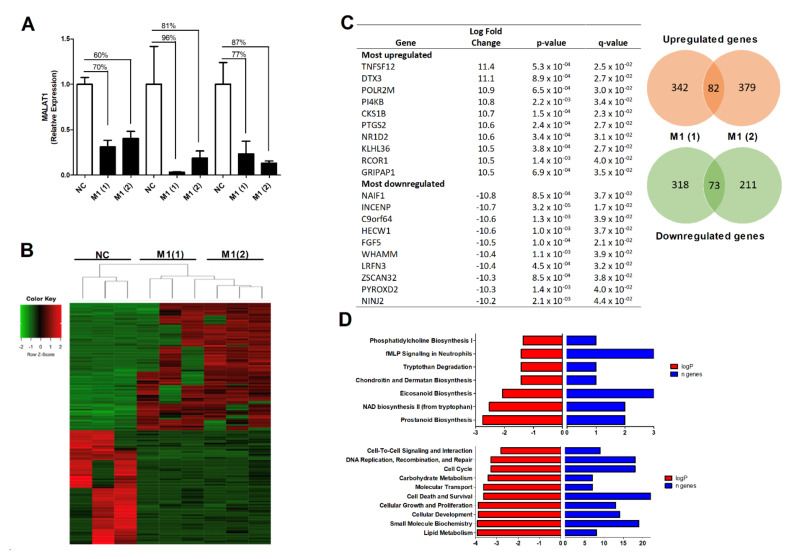
The effect of *MALAT1* knockdown on the OA osteoblast transcriptome. (**A**) Knockdown of *MALAT1* expression in primary OA osteoblasts. Osteoblasts were transfected with either a non-targeting control LNA (NC) or one of two LNAs targeting *MALAT1*, M1(1) and M1(2). Expression of *MALAT1* was determined by qPCR 24 h after transfection and normalised to 18S. (**B**) Heatmap of differentially expressed genes (>1.5-fold, *p* < 0.05) as determined by RNAseq (Quantseq) following 24 h LNA-mediated *MALAT1* knockdown. (**C**) Table of the most upregulated and down-regulated genes following *MALAT1* knockdown, with Venn diagram illustrating the total numbers of genes differentially expressed (>1.5-fold, *p* < 0.05) with each of the *MALAT1* LNAs. Values represent the mean fold change between control LNA and *MALAT1*(1) LNA and between control LNA and *MALAT1*(2) LNA. (**D**) Top canonical pathways and cellular processes as determined by Ingenuity Pathway Analysis of the differentially expressed genes (fold change > 1.5, *p* < 0.05) following *MALAT1* knockdown, with logP values representing the significance between the pathway/process and the gene dataset and n representing the numbers of genes within the dataset aligned to the pathway/process.

**Figure 3 cells-10-00786-f003:**
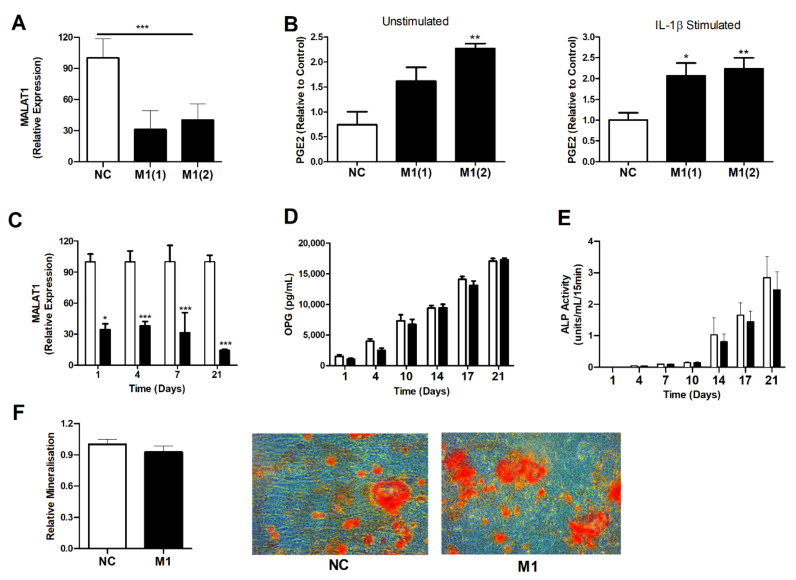
The effect of *MALAT1* knockdown on the function of OA osteoblasts. (**A**) Knockdown of *MALAT1* in OA osteoblasts (*n* = 6) following 24 h transfection with LNAs targeting *MALAT1* (M1(1) and M1(2)), compared to LNA control (NC). (**B**) Effect of *MALAT1* knockdown on 24 h basal and IL-1β mediated production of PGE2, as measured by ELISA. ** = *p* < 0.01 compared to non-stimulated control (*n* = 6). (**C**) Sustained knockdown of *MALAT1* expression in OA osteoblasts over 21 days following repeated transfection with a *MALAT1* LNA or control LNA. * = *p* < 0.05, *** = *p* < 0.001 compared to non-stimulated control (*n* = 6). (**D**) Production of OPG (pg/mL) over 21 days in control LNA and *MALAT1* LNA transfected OA osteoblasts (*n* = 6) as determined by ELISA. (**E**) Alkaline phosphatase (ALP) activity over 21 days in control LNA and *MALAT1* LNA transfected OA osteoblast lysates (*n* = 6). (**F**) Relative amount of osteoblast mineralisation following 21 days of transfection with control (NC) or *MALAT1* (M1) LNA as determined quantification of absorbance at following alizarin red staining, and representative light microscope images (×40 magnification) showing alizarin red stained mineralised bone nodules.

**Table 1 cells-10-00786-t001:** OA patient characteristics.

	All OA Patients	Knee OA Patients	Hip OA Patients
Age (years)	65 ± 2.2	67 ± 3.1	63.2 ± 3.2
Gender (m:f)	9:7	4:4	5:3
Height (cm)	167.2 ± 3.2	164.8 ± 5.4	169.4 ± 3.8
Weight (Kg)	82.4 ± 4.2	84.2 ± 7.9	80.8 ± 4.3
BMI (kg/m^2^)	29.7 ± 1.6	31.1 ± 2.7	28.4 ± 1.9
% Fat	34.8 ± 2.4	35.0 ± 5.1	34.8 ± 2.3
WHR ^1^	0.92 ± 0.02	0.93 ± 0.03	0.91 ± 0.03
KL Grade	4 (3.5–4)	4 (3.25–4)	4 (3.5–4)

^1^ WHR = Ratio of waist to hip circumference. All values represent mean ± SEM, except KL grade which is shown as median (25th–75th percentile).

**Table 2 cells-10-00786-t002:** Relationship between MALAT1 subchondral bone expression and the concentration of cytokines in OA patient serum.

	Concentration ^a^	r^2^	*p*-Value
Visfatin	2979 ± 355	0.11	0.23
Resistin	15441 ± 1394	0.17	0.18
Leptin	18365 ± 5865	0.03	0.55
Adiponectin	10640 ± 2081 ^b^	<0.01	0.94
Chemerin	6920 ± 937	<0.01	0.67
Dkk1	3.7 ± 0.3 ^b^	0.3	*0.04*
Galectin1	41 ± 3 ^b^	0.44	*0.009*
Eotaxin	170 ± 44	0.15	0.16
Amphiregulin	570 ± 54	0.24	0.07
Aggrecan	183 ± 54	0.10	0.28
FABP4	47.4 ± 15.2 ^b^	0.04	0.48
Serpin E1	131 ± 19 ^b^	0.02	0.63
IP10	32.3 ± 5.9	0.03	0.56
MIP1a	400 ± 76	0.06	0.39
MIP1b	188 ± 31	0.01	0.74
MCP1	398 ± 38	<0.01	0.94
MIP3a	37.3 ± 6.8	0.07	0.38
IL-1β	17.6 ± 3.1	0.007	0.78
IL6	10.9 ± 4.1	0.05	0.55
IL7	3.4 ± 0.3	0.09	0.29
IL10	4.5 ± 0.3	0.01	0.69
IL15	5.3 ± 1.6	0.02	0.66
TNF-α	4.3 ± 0.3	0.08	0.32
Gp130	94 ± 7.1 ^b^	0.02	0.63

^a^ Concentration of cytokines in pg/mL ± SEM. Except ^b^, concentration of cytokines in ng/mL.

## Data Availability

The data discussed in this publication have been deposited in NCBI’s Gene Expression Omnibus and are accessible through GEO Series accession number GSE167918. (https://www.ncbi.nlm.nih.gov/geo/query/acc.cgi?acc=GSE167918; deposited on 28 February 2021).

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
