# Peer review of "The Expression and Function of Metastases Associated Lung Adenocarcinoma Transcript-1 Long Non-Coding RNA in Subchondral Bone and Osteoblasts from Patients with Osteoarthritis"

_cells, 2021, doi:10.3390/cells10040786_

Round 1

Reviewer 1 Report

This is a well-designed and executed study.

There is a thorough review of existing literature although reconsider the reference to findings by Yang et al. 2018 (line 303) re. the regulation of OPG by MALAT-1. This finding was reported by Che et al. 2015 and consider difference in time frames.

Based on the error bars, Figure 3D seems to have significant differences in OPG levels at day 4 and day 17?

Typographical errors on lines 54, 65, 91, 94, 98, 99, 119, 166, 267, 271, 276.

Well done. 

Author Response

This is a well-designed and executed study.

Thank you for your review and constructive comments.

There is a thorough review of existing literature although reconsider the reference to findings by Yang et al. 2018 (line 303) re. the regulation of OPG by MALAT-1. This finding was reported by Che et al. 2015 and consider difference in time frames.

Thank you for this suggestion. We have replaced this reference in the discussion with the paper by Che et al. 2015.

Based on the error bars, Figure 3D seems to have significant differences in OPG levels at day 4 and day 17?

We performed a 2-way ANOVA with Bonferroni post-hoc tests to explore both the effect of treatment (NC vs MALAT1 knockdown) and time.  There was no effect of treatment (i.e MALAT1 KD), at any of the timepoints. But there was an effect of time on OPG production (i.e OPG production increased over time, irrespective of the treatment).

Typographical errors on lines 54, 65, 91, 94, 98, 99, 119, 166, 267, 271, 276.

Corrected. Thank you.

Well done.

Thank you for your review and constructive comments.

Reviewer 2 Report

In the manuscript Alnajjar et al., the authors investigated the association between LncRNA gene MALAT1 and osteoarthritis in OA patients. The authors performed MALAT1 knockdown experiments to determine its functional role in human AO osteoblasts. They claimed that expression of MALAT1 was significantly correlated with DKK1 and galectin-1, and that knockdown of MALAT1 significantly affected the downstream pathways associated with inflammatory mediation. The authors also carried out loss of function experiments to further determine the effect of MALAT1 on bone remodelling. In general, this manuscript is well written, the introduction is well structured and the method is, as far as I know, technically sound. I only have a few points that could make the analyses more robust.

1). For the expression analysis of MALAT1, the authors didn't mention its expression in control samples. It is important that control samples are included to determine the expression in both normal and OA samples. This applies to Figure 1A,B and C.

2). For the correlation analysis (Table 2), the authors didn't correct multiple testing. 24 tests were performed in total. It is highly likely that the significance of DKK1 (P = 0.04) will not survive after FDR correction.

3). All the gene names need to be italicised throughout the manuscript.

4). Line 250. I would be really careful to claim 'this is the first paper'. I would recommend rephrasing it.

Author Response

In the manuscript Alnajjar et al., the authors investigated the association between LncRNA gene MALAT1 and osteoarthritis in OA patients. The authors performed MALAT1 knockdown experiments to determine its functional role in human AO osteoblasts. They claimed that expression of MALAT1 was significantly correlated with DKK1 and galectin-1, and that knockdown of MALAT1 significantly affected the downstream pathways associated with inflammatory mediation. The authors also carried out loss of function experiments to further determine the effect of MALAT1 on bone remodelling. In general, this manuscript is well written, the introduction is well structured and the method is, as far as I know, technically sound. I only have a few points that could make the analyses more robust.

1). For the expression analysis of MALAT1, the authors didn't mention its expression in control samples. It is important that control samples are included to determine the expression in both normal and OA samples. This applies to Figure 1A,B and C.

Thank you for this suggestion. We have now performed comparative qPCR of MALAT1 expression in the subchondral bone of n=6 neck of femur control patients without OA and have updated Figure 1A accordingly.  Since these are patients without OA they are not assessed for OA severity via x-ray radiography. Therefore, we do not have clinical OA severity scores for them. Thus, Figure1B and 1C, which refer to MALAT1 expression with OA severity remain as before.

2). For the correlation analysis (Table 2), the authors didn't correct multiple testing. 24 tests were performed in total. It is highly likely that the significance of DKK1 (P = 0.04) will not survive after FDR correction.

We didn’t consider FDR correction as we considered it to be more appropriate to report the individual p-values, rather than risk a type II error where we report a false negative. FDR is usually performed to correct for multiple comparisons of genomics data where 1000s of genes are quantified. In this paper our analysis involved a relatively low number of measurements (24 cytokines). However, we do accept the reviewer’s point that a type I error cannot be ruled out. Therefore, we have now stated this clearly in the revised manuscript and that the findings were not significant upon FDR correction.

3). All the gene names need to be italicised throughout the manuscript.

Thank you. This is corrected in the revised manuscript.

4). Line 250. I would be really careful to claim 'this is the first paper'. I would recommend rephrasing it.

Thank you. This is now corrected in the revised manuscript.